# A New Mechanism of Carbon Metabolism and Acetic Acid Balance Regulated by CcpA

**DOI:** 10.3390/microorganisms11092303

**Published:** 2023-09-13

**Authors:** Yupeng Zhang, Fengxu Xiao, Liang Zhang, Zhongyang Ding, Guiyang Shi, Youran Li

**Affiliations:** 1Key Laboratory of Industrial Biotechnology, Ministry of Education, School of Biotechnology, Jiangnan University, Wuxi 214122, China; 7200201078@stu.jiangnan.edu.cn (Y.Z.); 8202306022@jiangnan.edu.cn (F.X.); zhangl@jiangnan.edu.cn (L.Z.); zyding@jiangnan.edu.cn (Z.D.); gyshi@jiangnan.edu.cn (G.S.); 2National Engineering Research Center for Cereal Fermentation and Food Biomanufacturing, Jiangnan University, 1800 Lihu Avenue, Wuxi 214122, China; 3Jiangsu Provincial Engineering Research Center for Bioactive Product Processing, Jiangnan University, Wuxi 214122, China

**Keywords:** glucose metabolic fluxes, CcpA, acetic acid metabolism, *Bacillus licheniformis*

## Abstract

Catabolite control protein A (CcpA) is a critical regulator in Gram-positive bacteria that orchestrates carbon metabolism by coordinating the utilization of different carbon sources. Although it has been widely proved that CcpA helps prioritize the utilization of glucose over other carbon sources, this global regulator’s precise mechanism of action remains unclear. In this study, a mutant *Bacillus licheniformis* deleted for CcpA was constructed. Cell growth, carbon utilization, metabolites and the transcription of key enzymes of the mutant strain were compared with that of the wild-type one. It was found that CcpA is involved in the regulation of glucose concentration metabolism in Bacillus. At the same time, CcpA regulates glucose metabolism by inhibiting acetic acid synthesis and pentose phosphate pathway key gene *zwF*. The conversion rate of acetic acid is increased by about 3.5 times after *ccpA* is deleted. The present study provides a new mechanism of carbon metabolism and acetic acid balance regulated by CcpA. On the one hand, this work deepens the understanding of the regulatory function of CcpA and provides a new view on the regulation of glucose metabolism. On the other hand, it is helpful to the transformation of *B. licheniformis* chassis microorganisms.

## 1. Introduction

Microorganisms require the uptake of carbon sources and other nutrients from their environment to maintain cell growth and metabolic equilibrium [1,2]. Glucose is a frequently preferred carbon source and serves as the primary energy source for most microorganisms [3]. Over time, microorganisms have evolved their metabolic pathways for glucose to meet metabolic demands and adapt to environmental changes [4]. Upon uptake via glucose transporters, glucose undergoes metabolism through two primary pathways, glycolysis and the pentose pathway, to generate energy. Initially, glucose was transformed into glucose-6-phosphate (G6P) [5]. G6P serves as a crucial juncture for entry into either glycolysis or the pentose phosphate pathway (PPP) [6,7]. Preceding research has established the rate-limiting stage of glycolysis in mammalian cells and other eubacteria [8,9]. However, the intricate molecular mechanisms governing glucose metabolic fluxes in microorganisms remain ambiguous.

In the process of microbial evolution, well-established regulatory mechanisms evolved to adapt to the changes in the environment [10]. Among them, the metabolic regulation system of carbon sources is an important way to maintain life activities [11,12]. Many metabolic regulatory proteins and regulatory networks have been reported [13,14]. As a model organism and engineering strain, *B. licheniformis* has a complete regulatory network of carbon metabolism [15,16]. CcpA is a global regulator and many metabolic pathways are regulated by CcpA including carbon sources metabolic system [17,18,19]. Previous studies have demonstrated a lot of mechanisms regulated by the CcpA protein such as carbon catabolic repression, the regulation of the β-hemolysin gene cluster, fructooligosaccharides metabolism and so on [20,21,22]. Glucose is the preferred carbon source and is regulated by CcpA for *B. licheniformis*, and the regulation of its metabolic flux and the regulation mechanism of the main overflow metabolites remain to be studied.

During glucose metabolism, a plethora of secondary metabolites are produced in addition to providing energy for cellular growth [23,24,25]. Acetic acid is a prominent overflow metabolite that is frequently secreted during microbial growth. On the one hand, the excessive production of acetic acid can lead to a decrease in pH in the culture medium and inhibit the growth of microorganisms; on the other hand, the existence of acetic acid can achieve a reasonable distribution of energy by regulating carbon metabolic flow. Previous studies have shown that the presence of acetic acid in the culture medium can inhibit glycolysis pathway genes and activate genes of TCA and ethanol oxidation-related enzymes. The metabolic pathways of acetate acid have been demonstrated in *B. subtilis* [26]. The key genes of the acetic acid metabolic pathway are *ptA* and *acK* and pyruvate is catalyzed by Ack and PtA to form acetic acid and the expression of *ackA* and *ptA* is repressed by CcpA by binding to the binding sites [26]. In the study of *Streptococcus mutans*, it was found that the acetic acid metabolism pathway gene *ptA* was repressed by CcpA [27]. On the other hand, acetic acid metabolism is also one of the important ways of microbial productivity to ensure the normal metabolic process of microorganisms [28,29]. A previous study has demonstrated that the acetic acid biosynthesis pathway gene is activated by CcpA [30]. At the same time, microorganisms have obtained a series of precise regulation mechanisms to balance acetic acid metabolism in the process of evolution [31]. But these precise mechanisms are not clear.

*B. licheniformis* has great application in industrial and experimental research [32,33]. *B. licheniformis* can quickly use glucose to provide energy for cell growth and produce overflow metabolites. Microorganisms can balance energy supply and overflow metabolism by self-regulation. A full understanding of its regulatory mechanism is helpful to the development and utilization of *B. licheniformis*. In this study, by analyzing the glucose consumption and related metabolic pathways of *B. licheniformis*, we found that CcpA regulates glucose metabolic flux by controlling the key genes of the pentose phosphate pathway (PPP). At the same time, CcpA can inhibit the expression of acetic acid metabolism pathway genes to achieve the balance between energy supply and overflow metabolism. The present study provides a new mechanism of carbon metabolism and acetic acid balance regulated by CcpA.

## 2. Results

### 2.1. The Metabolism of Single Carbon Source Containing Six Carbon Sugars Is Obviously Regulated by CcpA

Carbon metabolism is one of the main metabolisms of microorganisms. Glucose is an important carbon source in carbon metabolism. Previous studies have shown that CcpA plays an important role in the regulation of carbon metabolism, and the regulatory mechanism of glucose metabolism in *B. subtilis* has been proved [22,34]. *B. licheniformis* has a wider range of carbon substrates than other model microorganisms, so the regulation of carbon metabolism by CcpA may be different. In order to better study the regulatory mechanism of CcpA, we compared the use of common carbon sources (glucose, fructose, xylose, rhamnose, mannitol, sorbitol, glycerol and trehalose) after CcpA deletion, as shown in Figure 1 and Appendix A. According to the above results, we found that the use of glucose, mannitol and trehalose was significantly suppressed compared with the original control (Figure 1A–D). When cultured in the medium containing glucose (30 g/L), we obviously observed that the original bacteria could consume glucose in about 12 h, but in the same culture time (12 h), there was about 10 g/L left in the medium after CcpA knockout (Figure 1A). Similarly, in mannitol and trehalose consumption experiments, CcpA defective strains consumed only 50% when the original bacteria consumed mannitol and trehalose (Figure 1B,C). Similarly, when we compared the specific consumption rates of different carbon sources, we found that there were significant differences in glucose mannitol and trehalose after *ccpA* knockout (Figure 1D).

According to the above experimental results of carbon source consumption, we analyzed the significantly different carbon source catabolism pathways (Figure 1E). Interestingly, we found that the metabolic pathways of these carbon sources were highly similar to glucose metabolism. According to the above results, deletion of CcpA affects carbon metabolism, especially glucose-related metabolism. The specific regulatory mechanism of CcpA on glucose metabolism remains to be studied. Glucose is a preferred carbon source that can be quickly used to produce energy and metabolites in bacilli, including for the production of acetoin, 2,3-butanediol, acetic acid and so on [35]. A previous study has demonstrated that CcpA mutants can affect the use of other carbon sources and the synthesis of related metabolites in the presence of glucose [36,37,38]. In order to further research the regulation mechanism of CcpA on glucose utilization. We cultured *B. licheniformis* and *B. licheniformis CA* in a medium containing different concentrations of glucose. The growth and glucose utilization were monitored. As shown in Figure 2, we cultured the original bacteria and *B. licheniformis CA* with two concentrations of glucose and supplemented with glucose 24 h later and found that there was no significant difference in the growth of the two bacteria at the concentration of 20 g/L, but the OD_600_ of the original bacteria increased significantly when the concentration of glucose increased to 60 g/L. Subsequently, in order to further verify that the above growth differences were caused by the regulation of glucose metabolism by CcpA, we cultured *B. licheniformis* and *ccpA* defective strains in the medium without glucose, and we found that there was no difference in the growth of *B. licheniformis* and *ccpA* defective strains without glucose (Figure 2B).

Then, we found an interesting phenomenon by detecting the glucose content in the culture medium. Compared with the original bacteria, glucose consumption was significantly inhibited when *ccpA* was deleted. Under the condition of 20 g/L glucose culture, the consumption of glucose was not obvious. However, under the condition of 60 g/L glucose culture, the glucose of *B. licheniformis CA* was about 45 g/L when the glucose consumption of the original bacteria was finished. Similarly, after adding glucose for 24 h, the original bacteria could consume quickly, but the glucose content of *B. licheniformis CA* did not decrease after adding glucose (Figure 2C). Furthermore, we analyzed the specific consumption rate in the first 12 h. As shown in Figure 2D, when the glucose content was 20 g/L, there was no significant difference in the specific consumption rate, but when the glucose content was 60 g/L, the maximum specific consumption rate of *B. licheniformis CA* was about 3 times lower than that of the original bacteria. The above results showed that *B. licheniformis* had a certain inhibitory effect on glucose metabolism after *ccpA* deletion. Previous reports have shown that CcpA can activate certain genes in the glycolysis pathway [39]. Our results are consistent with previous reports that CcpA plays an important role in the regulation of glucose metabolism.

What is interesting is that the disorder of glucose metabolism after ccpA deletion is related to the glucose content in the culture medium. According to the above results, the maximum consumption of glucose is about 20 g/L. In order to verify the above conjecture, we cultured *B. licheniformis* and *B. licheniformis CA* in a medium containing different contents of glucose (0 g/L, 10 g/L, 20 g/L and 50 g/L). We detected growth and the amount of glucose remaining in the culture medium, as shown in Figure 3. Similar to the culture conditions without glucose, with the addition of 10 g/L glucose, the deletion of the CcpA protein coding gene did not have a statistical difference in growth and utilization (Figure 3A,B). Interestingly, with the increase in glucose content in the culture medium, there was a significant difference in glucose consumption rate when there was no significant difference in growth level. As shown in Figure 3B, compared with the original bacteria, the time for *B. licheniformis CA* to finish using 20 g/L glucose was delayed by 6 h (Figure 3B). However, there was no statistical difference in growth between the two groups (Figure 3A). Similarly, with the increase in glucose content to 50 g/L, the original bacteria could be consumed in about 15 h, but for *B. licheniformis CA*, glucose was no longer consumed when glucose was consumed to 30 g/L. The above results showed that the glucose metabolism of *B. licheniformis* was significantly inhibited with the increase in glucose content in the culture medium after *ccpA* deletion. The above results show that CcpA can reasonably regulate the metabolic process according to different concentrations of glucose and ensure the glucose utilization of microorganisms.

### 2.2. CcpA Not Only Activates Glycolysis but Also Inhibits Pentose Phosphate Pathway

To understand more clearly the regulation of glucose metabolism by CcpA, we first analyzed the glucose metabolism pathway of *B. licheniformis*. Through genome mining of *B. licheniformis*, we identified several key genes in the glucose metabolism pathway of *B. licheniformis*, as shown in Figure 4A (*glcU, ptsG, gdH, zwF, glcK, pjI* and *pyK*). There are two main pathways of glucose metabolism in *B. licheniformis*: glycolysis and pentose phosphorylation pathway (PPP). *glcU* and *ptsG* are mainly responsible for glucose transport and phosphorylation. *zwF*, *gdH* and *glcK* are node genes in the process of glycolysis and the *p* pentose phosphorylation pathway (PPP) in glucose metabolism. Then, we detected the transcriptional level of glucose metabolism pathway genes of *B. licheniformis* and *B. licheniformis CA* cultured in a medium with different glucose contents (0 g/L, 20 g/L and 60 g/L). The transcriptional results showed that the transcriptional intensity of the key gene (*zwF*) of the pentose phosphate pathway increased in different culture periods after *ccpA* knockout (Figure 4B–E). Meanwhile, the intracellular amount of glucose of *B. licheniformis CA* was about four folds higher than that of *B. licheniformis* and the intracellular content was maintained at a certain concentration, as shown in Figure 4F. Therefore, the deletion of ccpA may lead to changes in other metabolic regulatory processes, resulting in the use of glucose.

### 2.3. The Presence of CcpA Can Significantly Inhibit the Metabolism and Synthesis of Acetic Acid

Previous studies have shown that when microorganisms are cultured in the medium with glucose as substrate, they will produce overflow metabolites such as acetic acid, and the synthesis of acetic acid will lead to a low pH in the medium and inhibit the growth of microorganisms. Based on the above experimental results, we found that the use of glucose was inhibited when the glucose content in the culture medium was higher than that of 20 g/L after *ccpA* was knockout, so we detected the transcriptional level of acetic acid metabolic pathway and the content of acetic acid under different concentrations of glucose. As shown in Figure 5B, after *ccpA* was deleted, the transcriptional intensity of the acetic acid metabolism pathway genes *ackA* and *ptA* of *B.licheniformis* significantly increased. After 9 h of culture, under the condition of medium glucose concentration of 20 g/L, the transcriptional levels of *ackA* and *ptA* increased by 20 and 40 folds, respectively (Figure 5B). When the medium glucose concentration was 60 g/L, although the transcriptional level of *ackA* did not increase significantly, the transcriptional level of *ptA* increased by about 5 folds. When the culture time was extended to 24 h, because 20 g/L glucose had been consumed (Figure 5C), the transcription levels of *ackA* and *ptA* were not significantly different from those of the original bacteria under the culture condition of 20 g/L glucose, but the transcription levels of *ackA* and *ptA* were increased by 4–6 folds under the culture condition of 60 g/L glucose (Figure 5B).

Similarly, we detected the acetic acid content in the culture medium, and compared with the original bacteria, the loss of *ccpA* led to a significant increase in acetic acid content in the culture medium (Figure 5C). In addition, through the calculation of acetic acid conversion, we found that the acetic acid conversion increased to more than 50% after *ccpA* was knockout (Figure 5D). According to the above results, CcpA inhibits the metabolism of acetic acid synthesis and ensures the normal metabolism of glucose to provide energy for cell growth.

### 2.4. Molecular Mechanism of Glucose Metabolic Flux Regulated by CcpA

CcpA exerts its regulatory function mainly by binding to specific sites [40,41]. To better understand the molecular mechanism of CcpA regulating glucose metabolic flux, we used EMSA to validate the promoter regions of key genes in the glucose metabolism pathway. As shown in Figure 6A,B, with the increase in CcpA content in the reaction system, we found that the migration of fragments was significantly blocked. Subsequently, we analyzed the promoter sequences *gdH* and *zwF* and found that there were cre sites in these two promoters (Figure 6C). From the above experimental results, we found that CcpA of *B. licheniformis* regulates glucose metabolic flow by controlling the key genes of glycolysis and the pentose phosphate pathway during glucose metabolism, and can inhibit the synthesis of the overflow metabolite acetic acid to maximize energy utilization.

## 3. Discussion

Glucose is one of the most common carbon sources that can be used by microorganisms. Glucose provides energy for cell growth and produces secondary metabolites through glucose metabolism [42]. The pathway of glucose metabolism of microorganisms is complex and regulated by a variety of regulatory factors [43]. For the transformation and commercial use of chassis microorganisms, a thorough understanding of how microorganisms regulate their glucose metabolism is crucial. Microorganisms have a variety of metabolic mechanisms for metabolizing glucose. Transcription factors play a critical part in the metabolism of glucose, which is initiated when glucose enters the microbe and is carried out by a number of metabolic pathways that work in concert to ensure its smooth progression. The balance of microbial metabolism and energy is maintained by this intricate metabolic regulatory mechanism. Previous research has demonstrated that CcpA activates the *B. subtilis* glycolysis pathway and acetic acid metabolism pathway, but does not control the pentose phosphate pathway. We discovered a novel CcpA regulatory mechanism in *B. licheniformis* that differs from that in *B. subtilis* [30]. According to this study, CcpA inhibits the genes *zwF*, which is involved in the pentose phosphate pathway, and *ackA* and *ptA*, which are involved in the metabolism of acetic acid. It is crucial for maintaining the equilibrium of bacteria’ internal energy metabolism. This work offers a fresh perspective on the investigation of glucose metabolism.

In the process of evolution, Bacillus evolved different regulatory systems to cope with the complex environment [44]. Microorganisms modify the intensity of the glucose metabolism pathway through regulatory mechanisms in response to the various quantities of glucose in the environment. Microorganisms have the ability to develop quickly and take over the environment through population dominance, ensuring the quick utilization of the environment’s carbon supplies. CcpA is a recognized regulator of carbon metabolism, and Bacillus has numerous regulatory components that regulate the metabolic process in common [34]. Both single carbon sources and mixed carbon sources are subject to CcpA regulation. Our findings indicate that CcpA is capable of adjusting to variations in the environment’s glucose concentration and balancing the metabolic process to ensure the proper metabolism of bacteria.

Overflow metabolism is another crucial process for the metabolism of glucose in bacteria, along with glycolysis and pentose phosphate synthesis. One of the most prevalent overflow metabolites, acetic acid, has received a lot of attention [45]. Acetic acid can balance the central metabolism when present while an excess can prevent the growth of bacteria [46]. As a result, microbes have evolved sophisticated regulatory mechanisms to deal with complex surroundings [47]. Among them, the Pta-Acka system is recognized as the metabolic pathway of acetic acid synthesis in Bacillus [26]. We discovered that acetic acid content significantly increased when the CcpA protein was lost. Genes involved in the metabolism of acetic acid were also significantly activated. The production of acetic acid also inhibits the use of glucose at the same time. It is possible that too much acetic acid causes the pH to drop, which stops microbes from growing.

The above experimental results can be used for reference. First off, the deletion of *ccpA* causes the pentose phosphate pathway genes in the glucose metabolism route to become active, which in turn causes the glucose metabolism flux to be regulated. However, the degree of this regulation depends on how much glucose flows through each pathway. This work suggests that CcpA serves as a glucose concentration sensing component since its absence causes a malfunction in how Bacillus regulates glucose concentration. There are numerous ways to control the synthesis and metabolism of acetic acid at the same time. According to research, CcpA is required to initiate acetic acid synthesis in *B. subtilis*; however, *Streptococcus* mutans have a different acetic acid production metabolism [26,48,49]. The acetic acid synthesis pathway is activated in this study when the CcpA protein is deleted, which may be related to the different cre sites in the acetic acid synthesis genes. The precise mechanism, however, needs more research.

## 4. Conclusions

The most fundamental metabolic activity in bacteria is glucose metabolism. The thorough study of glucose metabolism is crucial for the controlled evolution and use of microorganisms. CcpA is a universal regulator that is involved in a variety of metabolic activities. In this work, we compared the changes in glucose uptake, the transcription of relevant metabolic pathways, and overflow metabolites after *ccpA* deletion. According to the aforementioned findings, CcpA can be used as a protein that regulates glucose metabolism because it can prevent the pentose phosphate pathway and ensure that microorganisms have normal central metabolisms in a range of glucose concentrations. CcpA can also inhibit acetic acid synthesis to a certain extent and promote the growth of microorganisms.

## 5. Materials and Methods

Bacterial strains and reagents. Table 1 includes a list of the plasmids and strains used in this work. The *E. coli* JM109 strain clones every shuttle plasmid. In LB medium (10 g/L tryptone, 5 g/L yeast extract, and 5 g/L NaCl) at 37 °C, 200 rpm, *E. coli* JM109 and DE3 were grown. In LB medium and TB media (12 g/L tryptone, 24 g/L yeast extract, 12.54 g/L K2HPO4, and 2.31 g/L KH2PO4) at 37 °C, 250 rpm, *B.licheniformis* were grown. Tetracycline (20 g/mL), kanamycin (30 g/mL), and spectinomycin ampicillin (100 g/mL) were added to the growth media as supplements. In the TB medium, *B.licheniformis* growth and glucose uptake are seen at various glucose concentrations. TB medium was utilized to cultivate the *E. coli* DE3 used for CcpA expression. The medium was added with IPTG when the culture’s OD_600_ ranged from 0.6 to 0.8. Sinopharm (Sinopharm Chemical Reagent, Shanghai, China) is where all reagents are acquired.

Plasmids and strains construction. All experimental operations are strictly in accordance with the guidelines for molecular cloning experiments. *B. licheniformis CA* is a strain constructed by knocking out the *ccpA* on the basis of *B. licheniformis*. The method of *ccpA* gene knockout is carried out according to the method we reported earlier [36]. pET28a-CcpA was the plasmid used for the expression of CcpA. *ccpA* was cloned from the genome. The sequence of *ccpA* was verified by sequencing. The constructed plasmid was transformed into *E. coli* DE3 by chemical transformation to construct recombinant bacteria.

Growth and glucose consumption detection. First, *B. licheniformis* and *B. licheniformis CA* were activated on the plate, and then the single colony was cultured in an LB medium for 16 h as seed liquid. Then, the seed liquid was inoculated into TB medium according to a 3% inoculation amount and different concentrations of glucose were supplemented. Then, the inoculated TB medium was cultured at 37 °C 250 rpm. The absorbance at 600 nm was measured at intervals of 3 h. At the same time, the 1 mL sample was centrifuged at 12,000 rpm for 5 min, and the supernatant was precipitated overnight by adding 10% trichloroacetic acid of the same volume. Then, 12,000 rpm was centrifuged for 20 min, and the supernatant was taken into the injection bottle. HPLC (Thermo Fisher Scientific, Waltham, MA, USA) with a refractive index detector and chromatographic column was used to detect the consumption of sugars. The column temperature was maintained at 80 °C and the mobile phase was H_2_O running at 0.6 mL/min.

Detection of the intracellular amount of glucose. *B. licheniformis* and *B. licheniformis CA* were cultured in a TB medium with glucose. Then, 1 mL culture was taken at 12 h and 24 h, the supernatant was centrifuged at 12,000 rpm, and the cells were washed twice with 0.1 mM pbs buffer. The washed cells were transferred to the weighed 1.5 mL centrifuge tube to weigh their wet weight. Then, resuscitated the cells with 0.1 g/L lysozyme solution and fixed the volume to 1 mL, reacted for 30 min at 37 °C, and then ultrasonic crushed the cells. After 12,000 rpm centrifugation, 500 μL of supernatant was added to 500 μL 10% trichloroacetic acid overnight to precipitate and remove proteins. Then, the content in the sample was determined by the liquid phase.

RNA extraction and qRT-PCR. *B. licheniformis* and *B. licheniformis CA* were raised in a TB medium that had varied amounts of glucose added. Every three hours, samples were obtained. According to the directions, the culture was centrifuged for 12,000 rpm to collect the bacteria, then washed twice with ddH_2_O before being resuspended with 100 mg/mL lysozyme. After the reaction had been going on for 15 min at room temperature, 100 L of R1 was added, and after another 500 L of R2 had been added and allowed to react for 5 min, the supernatant was centrifuged in an adsorption column, twice washed with washing buffer, and then eluted with RNA-free water. Using the Quawell Q5000, the samples’ RNA content was calculated. The cDNA was created from the RNA using a cDNA synthesis kit from Vazyme Biotech in Nanjing, China. SYBR green was used for qRT-PCR in a CF96 Real-time machine (qTOWER3G IVD). All primers used for the qRT-PCR are listed in Table 2.

EMSAs. Electrophoretic mobility shift assays (EMSAs) were performed according to the direction of EMSA. Briefly, *E. coil* DE3 was used for the heterologous expression of CcpA used in the research. Subsequently, CcpA was purified using Mag-beads His-tag Protein Purification (Sangon Biotech, Shanghai, China). Saline ions in eluent were removed by dialysis. Labeled fragments used were amplified with the primers listed in Table 2. First of all, according to the operating instructions, the binding buffer was mixed with CcpA protein and left at room temperature for 10 min to eliminate non-specific binding. Then, the biotin-labeled fragments were added to the above reaction system. After the reaction at room temperature for 20 min, electrophoresis was carried out. After the end of electrophoresis, the electrophoretic model was transferred to the nylon film with a positive side. Then, the purple diplomatic connection was carried out and the sealing liquid was sealed. Finally, it was developed under the gel imager.

## Figures and Tables

**Figure 1 microorganisms-11-02303-f001:**
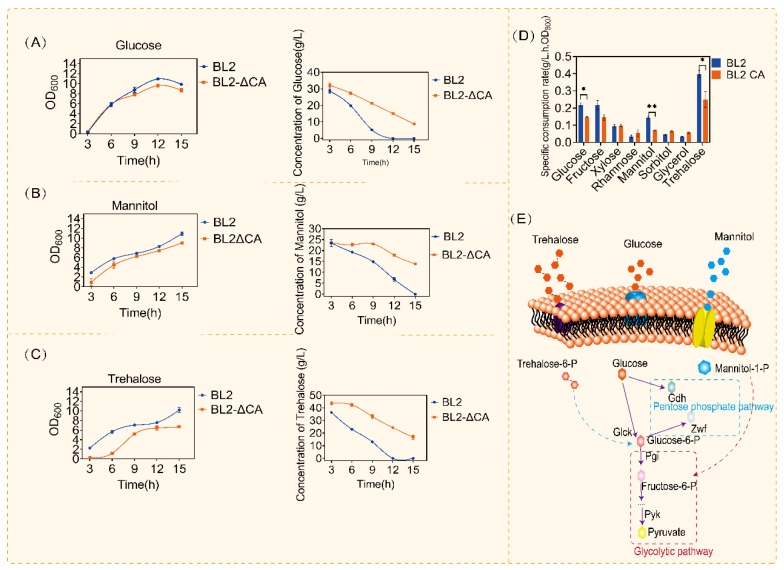
Comparison of consumption of different carbon sources by CcpA defective strains. (**A**) Glucose consumption and OD_600_ of *Bacillus licheniformis* and *Bacillus licheniformis CA* in medium with glucose. (**B**) Mannitol consumption and OD_600_ of *Bacillus licheniformis* and *Bacillus licheniformis CA* in medium with mannitol. (**C**) Trehalose consumption and OD_600_ of *Bacillus licheniformis* and *Bacillus licheniformis CA* in medium with trehalose. (**D**) Metabolic pathways of glucose, mannitol and trehalose. Statistical comparisons were performed using two-sided Student’s *t*-test (* *p* < 0.05, ** *p* < 0.01). (**E**) Metabolic pathways of glucose, mannitol and trehalose.

**Figure 2 microorganisms-11-02303-f002:**
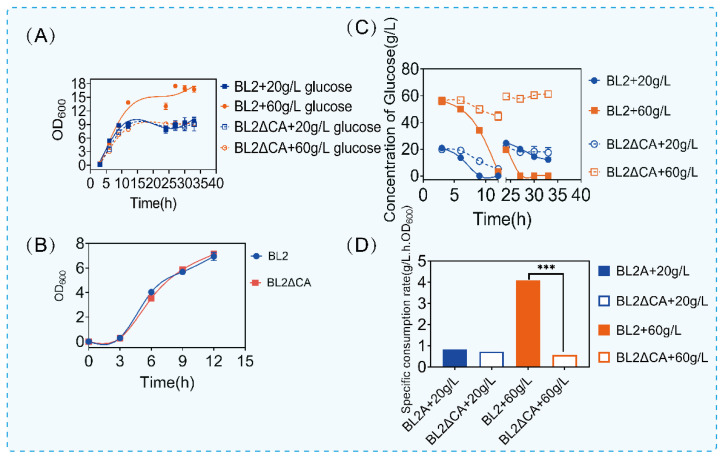
Growth and glucose consumption of *Bacillus licheniformis* and *Bacillus licheniformis CA*. (**A**) The growth of *Bacillus licheniformis* and *Bacillus licheniformis CA* cultured with 20 g/L and 60 g/L glucose. (**B**) The growth of *Bacillus licheniformis* and *Bacillus licheniformis CA* cultured with TB medium. (**C**) Glucose content in culture medium at different times. (**D**) Maximum specific glucose consumption rate of *Bacillus licheniformis* and *Bacillus licheniformis CA* under different glucose content culture conditions. Statistical comparisons were performed using two-sided Student’s *t*-test (*** *p* < 0.001).

**Figure 3 microorganisms-11-02303-f003:**
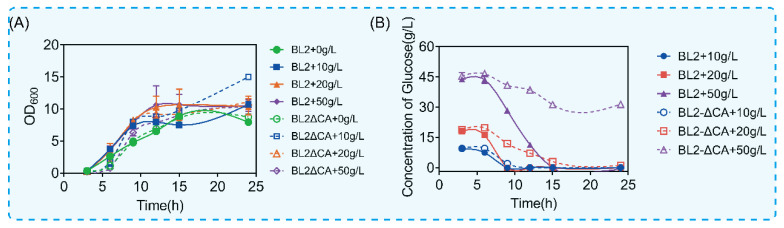
Growth and glucose consumption under culture conditions with different concentrations of glucose. (**A**) Growth of glucose under different glucose concentrations. (**B**) Glucose content in culture medium at different times.

**Figure 4 microorganisms-11-02303-f004:**
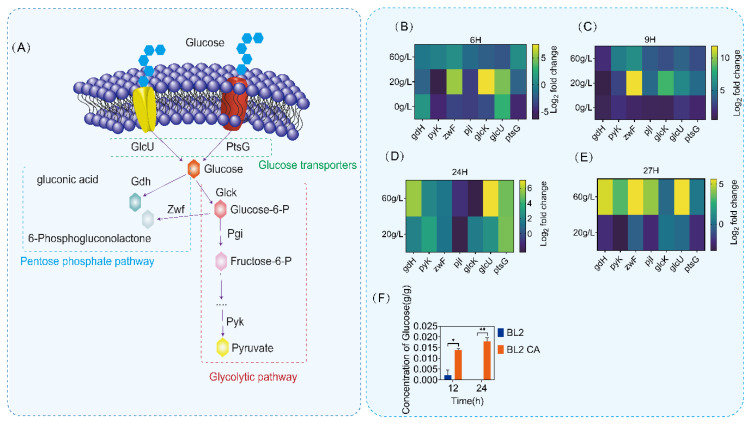
Determination of glucose metabolic pathway and transcriptional intensity. (**A**) Identification of glucose metabolic pathway (glycolysis pathway and pentose phosphate pathway). The pentose phosphate pathway has a blue border, and the pentose phosphate pathway has a red border. (**B**–**E**) Determination of transcriptional intensity of glucose metabolic pathway in 6 h, 9 h, 24 h and 27 h. (**F**) The detection of intracellular glucose content in different time periods showed that blue was primitive bacteria and orange was *Bacillus licheniformis CA*. Statistical comparisons were performed using two-sided Student’s *t*-test (* *p* < 0.05, ** *p* < 0.01).

**Figure 5 microorganisms-11-02303-f005:**
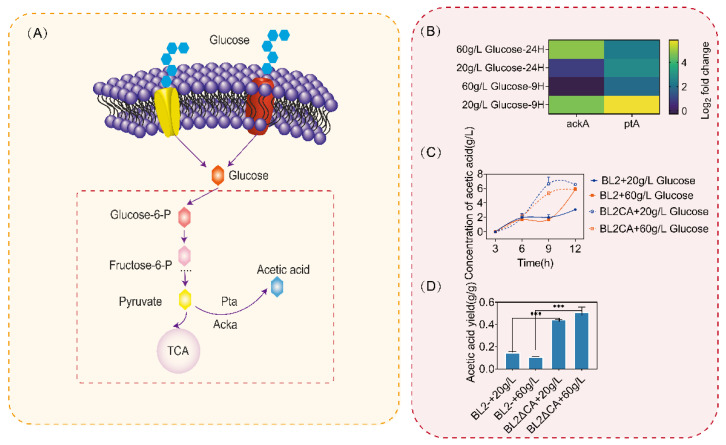
Acetic acid metabolic pathway and acetic acid synthesis. (**A**) Acetic acid metabolism pathway of *Bacillus licheniformis*. (**B**) Determination of transcriptional intensity of acetic acid metabolic pathway. (**C**) The content of acetic acid in the culture medium was determined. (**D**) Conversion rate of glucose to acetic acid of *Bacillus licheniformis* and *Bacillus licheniformis CA* cultured with different concentrations of glucose at 12 h. Statistical comparisons were performed using two-sided Student’s *t*-test (*** *p* < 0.001).

**Figure 6 microorganisms-11-02303-f006:**
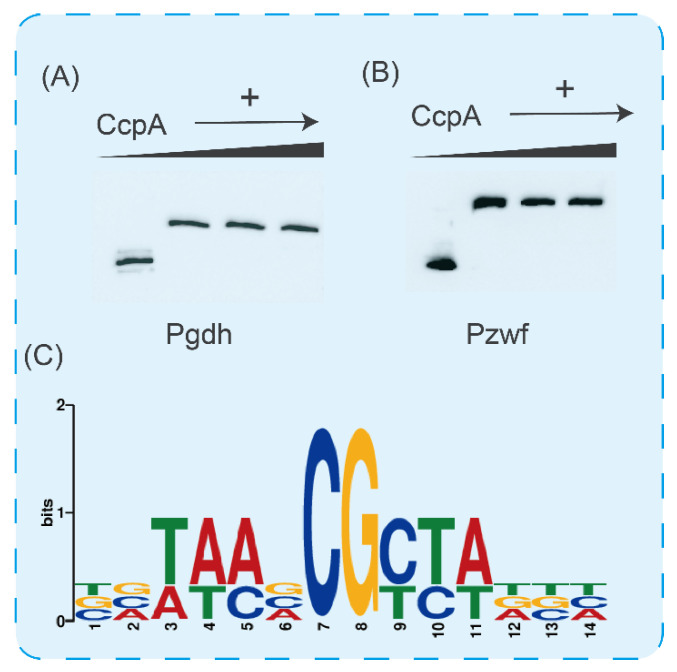
Study on the regulation mechanism of CcpA. (**A**) Verification of binding ability of CcpA and *gdH* promoter. (**B**) Verification of binding ability of CcpA and *zwF* promoter. Analysis of cre sites of *ptA* and *ackA* promoter in acetic acid synthesis pathway. (**C**) Conservative analysis of cre sites.

**Table 1 microorganisms-11-02303-t001:** Strains used in this study.

Strain	Description	Reference
Strains		
*Escherichia coli* JM109	F′, traD36, proAB + lacIq, Δ(lacZ), M15/Δ (lac-proAB), gln V44, e14−, gyrA96, recA1, relA1, endA1, thi, hsdR17 (CICIM B0012)	CICIM-CU
*E. coli* BL21(DE3)	F-ompT gal dcm lon hsdSB (rB-mB) λ(DE3)	CICIM-CU
*Bacillus licheniformis* CICIM B1391	wild-type (CICIM B1391)	CICIM-CU
*Bacillus licheniformis CA*	*B. licheniformis* CICIM B1391, Δ*ccpA*	CICIM-CU
BL21pETPA	BL21, harboring pET28aPA	CICIM-CU

**Table 2 microorganisms-11-02303-t002:** Plasmids used in this study.

Primers	Sequence
PtsG-Q-F	tacgaatcaggcgggagaaattgtc
PtsG-Q-R	aacccataataccggcaacaagc
GlcU-Q-F	aatcagctgaaaagcatcaagttgatcg
GlcU-Q-R	ttccagcgatgtcagcacgatt
GlcK-Q-F	gcggcattattgtaaacggagaaatc
GlcK-Q-R	cgcaagtctgtatctagatgaccgg
Pji-Q-F	gcgaaacgccgcatttatgc
Pji-Q-R	tcatttcatcgatatcggctccgc
ZwF-Q-F	aacaggggatttggcaaaacgaa
ZwF-Q-R	gtgagatgtaaactcgtccacatcct
Pyk-Q-F	tccttcttgatacgaaaggtccgg
Pyk-Q-R	ccatcgtcaagcaaaatcgttgaacc
Gdh-Q-F	gtaccttctgaggatttgtctctggaag
Gdh-Q-R	ttgcttgccgcataatggacaaaat
RpsE-F	tggtcgtcgtttccgcttcg
RpsE-R	tcgcttctggtacttcttgtgctt
ackA-Q-F	acaccgcgtcgtacacg
ackA-Q-R	gcattgtttggtggaatgccg
pta-Q-F	cgtatttgttttcccaagccttgaag
pta-Q-R	caaaaacaaattacagcgcttgaactgc
Pgdh-F-Bio	ttcttatgtactccctccataaccgct
Pgdh-R	aaaagaaggcggggtgccttc
Pzwf-Bio-F	atatcctttcctcctgctaaaaacttcatctatc
Pzwf-R	attaaacgtacctcactttattcgaagctaagt

## Data Availability

There are no new data were created.

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
