# Peer review of "A New Mechanism of Carbon Metabolism and Acetic Acid Balance Regulated by CcpA"

_microorganisms, 2023, doi:10.3390/microorganisms11092303_

Round 1

Reviewer 1 Report

Dear Authors,

The research work performed in this study is interesting and scientifically sound. However, there are some flaws which need to be addressed as per suggested in comments section.

Author Response

Comments 1. Fig.1. Contains lot of figures which are too small in size and it is not clearly visible

particularly titles on X-and Y-axis. Overall, mysuggeststion is to improve the font

and figure sizes so as to get clear visibility.

Response 1:

Thanks for your comments. We have adjusted the size and the clarity of the picture. The main figures have been rearranged and the rest of the figures have been added to the supplementary material.

Comments 2. Also,Fig. 4is not clear, Please improve the size and resolution of figure 4.

Response 2:

Thanks for your comments. We have adjusted the size and the clarity of the picture.

Comments 3. There are several articles have been published on the topic related acetic acid production with higher productivities and yields. Why do you think this study will have significance over reported studies?

Response 3:

Thanks for your comments. The synthesis of acetic acid and the control of associated genes have been documented in earlier investigations. Acetic acid, a typical overflow metabolite, can also limit cell development at the same time. There is still research to be done on the precise regulating mechanism of overflow metabolism as well as how to balance the growth energy supply of microorganisms while employing carbon sources. CcpA is a key player in numerous metabolic pathways of bacteria as a global regulator. In this work, it was discovered that the deletion of CcpA protein would result in a large increase in the rate of acetic acid conversion and a significant reduction of Bacillus licheniformis growth in a glucose-containing environment. Thus, we discovered a novel and intriguing mechanism: the presence of the CcpA protein can effectively balance the metabolism of acetic acid and the supply of growth energy, creating an environment that is favorable for cell growth.

Comments 4. Please improve the font sizes in all figures

Response 4:

Thanks for your comments. The font size of all figures have been improved.

Reviewer 2 Report

The manuscript reports a new mechanism of carbon metabolism and acetic acid bal- ance regulated by catabolite control protein A (CcpA). In particular, a mutant Bacillus licheniformis deleted for CcpA was constructed. Further, the changes in glucose uptake, the transcription of relevant metabolic pathways, and overflow metabolites after CcpA deletion

were studied. 

The work is interesting since the thorough study of glucose metabolism is crucial for the controlled evolution and use of microorganisms. The findings of the study suggest that CcpA can be used as a protein regulating glucose metabolism ensuring that microorganisms have normal central metabolisms in a range of glucose concentrations. 

English is in most of its parts well-written,  although minor revision is suggested.

Figures are too many. I suggest move some of the them to supplementary material.

Figure 6 is blurry and stretched; please fix it. 

Figure 4 caption. It should be reduced. 

Minor revision 

Author Response

Comments 1. Figures are too many. I suggest move some of the them to supplementary material.

Response1:

Thanks for your comments. Some figures of Figure 1 have been moved to supplementary material.

Comments 2. Figure 6 is blurry and stretched; please fix it.

Response 2:

Thanks for your comments. Figure 6 has been fixed.

Comments 3. Figure 4 caption. It should be reduced.

Response 3:

Thanks for your comments. Figure 4 caption has been reduced.
